# The Prevalence of Fetal Alcohol Spectrum Disorders in An American Indian Community

**DOI:** 10.3390/ijerph16122179

**Published:** 2019-06-20

**Authors:** Annika C. Montag, Rhonda Romero, Toni Jensen, Amiyonette Goodblanket, Ami Admire, Conner Whitten, Daniel Calac, Natacha Akshoomoff, Maria Sanchez, MarLa Zacarias, Jennifer A. Zellner, Miguel del Campo, Kenneth Lyons Jones, Christina D. Chambers

**Affiliations:** 1Department of Pediatrics, University of California San Diego, La Jolla, CA 92093-0828, USA; m3casillas@ucsd.edu (M.S.); mccastaneda@ucsd.edu (M.Z.); jazellner@ucsd.edu (J.A.Z.); midelcampo@ucsd.edu (M.d.C.); klyons@ucsd.edu (K.L.J.); chchambers@ucsd.edu (C.D.C.); 2Southern California Tribal Health Clinic; Connerftw@gmail.com (C.W.); 3Department of Psychiatry, University of California San Diego, La Jolla, CA 92093-0828, USA; nakshoomoff@ucsd.edu; 4Department of Family Medicine and Public Health, University of California San Diego, La Jolla, CA 92093-0828, USA

**Keywords:** fetal alcohol spectrum disorder, prevalence, American Indian Alaska Native

## Abstract

The prevalence of fetal alcohol spectrum disorders (FASD) differs among populations and is largely unknown among minority populations. Prevalence and characterization of FASD is necessary for prevention efforts and allocation of resources for treatment and support. However, prevalence data are lacking, including among many minority populations. The aim of this study was to obtain an FASD prevalence estimate in a Southern California American Indian community employing active case-ascertainment. In 2016, American Indian children aged 5–7 years and their caregivers were recruited in collaboration with Southern California Tribal Health Clinic. Children were assessed using physical examinations and neurobehavioral testing. Parent or guardian interviews assessed child behavior and prenatal exposures including alcohol. Of 488 children identified as eligible to participate, 119 families consented and 94 completed assessments to allow a classification for FASD. Participating children (*n* = 94) were an average of 6.61 ± 0.91 years old and half were female. Most interviews were conducted with biological mothers (85.1%). Less than one third (29.8%) of mothers reported consuming any alcohol in pregnancy and 19.1% met study criteria for risky alcohol exposure prior to pregnancy recognition. Overall 20 children met criteria for FASD, resulting in an estimated minimum prevalence of 41.0 per 1000 (4.1%). No cases of fetal alcohol syndrome (FAS) were identified; 14 (70.0%) met criteria for alcohol related neuro- developmental disorder (ARND). Minimum prevalence estimates found in this sample are consistent with those noted in the general population.

## 1. Introduction

Accurate estimates of the types and overall prevalence of fetal alcohol spectrum disorders (FASD) among minority populations in the U.S. are lacking. This information is crucial to target clinical resources and maximize efficacy of prevention, treatment and support within individual communities.

Estimates of FASD prevalence have recently been reviewed by Roozen et al. (2016) [1] and Lange et al. (2017) [2]. These estimates ranged from a global rate of 7.7 per 1000 to varying specific population rates, including the particularly high prevalence of 113.2 per 1000 in selected locations in South Africa. A recent U.S. study used active case ascertainment in a cross-sectional sample of general population first-grade children residing in four communities [3]. Prevalence estimates in that study ranged from 11.0 per 1000 to 50.0 per 1000 (1.1–5.0%) using the most conservative method for calculation. Using less conservative, weighted prevalence estimates, this study further suggested that FASD may affect as many as one in ten first-grade school children. However, it is unknown to what extent these estimates are generalizable to other populations.

The heterogeneous American Indian Alaska Native (AIAN) populations of the U.S. differ substantially in patterns of alcohol consumption, use of contraception, as well as preferences, cultural practices and traditions [4,5,6,7,8,9]. For this reason, it is reasonable to assume that the prevalence of FASD might differ by specific AIAN population as well.

The impetus for the present study lay in the prioritization of FASD as a health issue by a specific AIAN community. This was in part due to previous studies in this community documenting vulnerability to having an alcohol-exposed pregnancy among approximately one third of women of childbearing age [10,11]. In response to this risk of prenatal alcohol exposure, local American Indian (AI) community members expressed an interest in establishing support mechanisms for families dealing with the sequelae of alcohol-exposed pregnancies. However, without access to diagnosis and a baseline prevalence estimate of FASD, efforts to determine the support-related needs and priorities for prevention and treatment in the community would be limited.

The present study was undertaken to estimate the prevalence of FASD among a reservation-based Southern California AI community and to identify families dealing with the effects of FASD to allow determination of how best to support them.

## 2. Materials and Methods

This study was designed as an ancillary special population study under the umbrella of a larger parent study, the National Institutes of Health (NIH)—National Institute on Alcohol Abuse and Alcoholism (NIAAA)-funded Collaboration on FASD Prevalence (CoFASP) study which estimated the regional prevalence of FASD in the U.S. [3]. The parent study used a cross-sectional design to sample children 5–7 years of age in regular first grade classrooms over two academic years in four regional communities in the U.S. Children were selected for the parent study through active case ascertainment in public and private elementary schools. Participating children and parents/guardians underwent multi-tiered evaluations, in most cases including a screening tier for growth deficiency, followed by a full evaluation for the key domains required for a classification of FASD. These included growth measurements, a dysmorphological evaluation for physical features of FASD, a neurobehavioral testing battery and a maternal or collateral interview regarding alcohol consumption in the index pregnancy. Details regarding the sampling strategies and methods for the parent study are described elsewhere [3].

The ancillary study in the selected AI community employed the same cross-sectional study design and evaluation of the key domains using the same tools with specific attention to cultural considerations. However, eligibility was defined as an AI child registered at the Southern California Tribal Health Clinic in the age-range of 5–7 years and did not utilize school-based ascertainment. Consequently, no teacher assessments of behavior were collected. In addition, for the ancillary study, no screening tier was used to select children for the full evaluation. All eligible consented children were offered assessment for growth, dysmorphology, neurobehavior and prenatal alcohol exposure. Data from the ancillary study were not included in the regional prevalence estimate from the parent study.

### 2.1. Sample Source

All AI children from 5 to 7 years of age and their caregivers were eligible to participate. Families were recruited in collaboration with a Southern California Tribal Health Clinic from two health clinic sites located in Southern California. Clinic membership rolls as of January 2016 were used to identify the eligible sample.

### 2.2. Ethics

This study was conducted in accordance with the Declaration of Helsinki, and the protocol was approved by University of California, San Diego (UCSD-#111082) and the Southern California Tribal Health Clinic (SCTHC-approved 7/7/2015) Institutional Review Boards (IRBs). All staff members completed human subjects’ protections training. All caregivers provided written informed consent and the subset of children who were seven years of age at the time of consent provided written assent through a process whereby the consent/assent forms were read aloud. The purpose of the study was described to children who were under the age of seven. Participants were provided incentives commensurate with time and/or travel required for participation. Caregivers received a summarized report of findings from their child’s evaluations as well as referrals for services and support where relevant.

Data from the present study belongs to the tribal entities from which the data were collected. The Tribal IRB reviewed and approved this manuscript for publication (4/30/2019). Cultural considerations affected the manner of recruitment, contact with participants, and structuring of recruitment awareness events. Local AI community members were involved as research staff and contributed to the design of study procedures. Where non-AI staff were involved in the study, they were educated regarding local etiquette and other relevant issues.

### 2.3. Recruitment

Letters inviting eligible families to participate were sent from the Southern California Tribal Health Clinic. Culturally congruent methods were used to raise awareness and to encourage participation. Flyers were posted in the clinics, local Tribal Halls, and in the community newsletter. Information was provided through table displays at local community events.

The research team included local AI trusted community members not just to provide cultural and logistic insights, and prevent cultural mishaps and misunderstandings, but to enable the collection of valid data reflecting the community. Local team members conducted all recruitment. They were also responsible for all scheduling, participant transportation, and the majority of mother and child related interviews. The neurobehavioral testing staff was augmented by training one local AI team member to conduct neurobehavioral exams. Team members were extensively trained and carefully monitored. As much as possible, all aspects of the study involved local research staff. Food was provided at each research event or participant assessment interaction as is culturally expected.

### 2.4. Assessment

Assessments took place on participating reservations. Study staff representing specific types of expertise from the parent CoFASP study regional site located in San Diego, California traveled to the various reservation sites to perform physical examinations and neurobehavioral testing. The assessments were the same standardized, age-appropriate assessments as used in the CoFASP study and used the same cut-off criteria as described in May and Chambers et al. (2018) [3]. As this study was community based, not school based, no teacher questionnaires were included. The following assessments were performed:

#### 2.4.1. Physical Examination by a Dysmorphologist

○Measurement of the child’s height, weight, and head circumference○Ranking on a Likert scale the smoothness of the child’s upper lip and thinness of the vermilion border of the upper lip○Measurement of the length of the palpebral fissures (eye openings)○Evaluation for heart murmur using a stethoscope○2D and 3D facial photographs

#### 2.4.2. Neurodevelopmental Assessments

○Differential Ability Scales (2nd Edition) (DAS-II) [12]. Cronbach’s alpha in our sample was 0.819.○NEPSY-II (NEuroPSYchological Assessment) Subtests (2nd Edition) including attention/executive functioning, language, and sensorimotor functioning [13,14,15]. Cronbach’s alpha in our sample was 0.788.○Beery-Buktenica Developmental Test of Visual-Motor Integration (6th Edition) (VMI) for graphomotor skills [16,17,18]○Bracken Basic Concept Scale–Revised (BBCS-R) [19] which evaluates academic achievement and basic concept development including letters, colors, numbers, sizes, comparisons, shapes, direction/position, and time/sequence

#### 2.4.3. Parent/Guardian Interview Instruments Regarding the Child

○Vineland Adaptive Behavior Scales (2nd Edition) (VABS) [20]. This well normed and widely used measure of adaptive functioning provides a measure of “real world” functioning of the child. Cronbach’s alpha in our sample was 0.720.○Child Behavior Checklist (CBCL) [21,22,23] which includes 100 problem behaviors. The CBCL is commonly used in both research and clinical application to obtain standardized rating of various aspects of behavioral, emotional, and social functioning of the child. Cronbach’s alpha in our sample was 0.909.

#### 2.4.4. Biological Mother or Collateral Caregiver Structured Interview Regarding the Biological Mother and Pregnancy Including Questions Regarding Prenatal Alcohol Exposure

○Maternal health questions including alcohol use prior to pregnancy recognition and during pregnancy○Cofactors of maternal risk including demographics, maternal nutrition, tobacco and recreational drug use

#### 2.4.5. Diagnostic Classification

Criteria for physical and neurobehavioral features were identical to those used in the parent CoFASP study and were based on revised Hoyme criteria [24] including the following criteria for Risky Maternal Alcohol Consumption: ○≥3 drinks per occasion on ≥2 occasions during pregnancy○≥6 drinks per week for ≥2 weeks during pregnancy○Self (or collateral caregiver) report of alcohol-related events during pregnancy

### 2.5. Classification of FASD

The study diagnostic team met in case conference on four occasions during the course of the study to review the physical examination data, neurobehavioral assessments, 2D facial photographs, and interview data to determine FASD category for children whose data were sufficiently complete to allow for classification. Children were classified as fetal alcohol syndrome (FAS), partial fetal alcohol syndrome (pFAS), alcohol related neurodevelopmental disorder (ARND) or no FASD.

### 2.6. Statistical Analysis

Prevalence was estimated using the eligible population invited to participate as the denominator and the total number of children classified with an FASD as the numerator.

## 3. Results

Of 488 eligible children and their mothers or guardians, 119 were recruited, and 94 completed assessments with sufficient data to allow a classification for FASD and inclusion in this analysis (Figure 1). Data from 94 physical exams, 80 maternal interviews and 14 collateral interviews, 94 CBCL, 94 VABS, 94 DASII, 92 NEPSY, 94 VMI, and 90 BBCS assessments are represented. As shown in Table 1, half of all participating children were female and their average age was 6.61 ± 0.91 years. The majority of mothers interviewed were cohabitating (71.3%) and had at least some college education (61.2%). Most interviews were conducted with biological mothers (85.1%). Women ranged from 24–49 years old at the time they were interviewed and were an average of 7.53 ± 5.33 weeks (range 1–39 weeks, median 6.5, mode 8) of gestation when they became aware of their pregnancy. Prior to being aware of their pregnancy but in the first trimester, nearly 30% of women consumed some alcohol and 19.1% of them met the study criteria for an alcohol-exposed pregnancy.

Prenatal exposures are shown in Table 2. Among the 28 women (29.8%) who consumed alcohol while pregnant before they were aware of their pregnancy, 66.7% drank at least once per week and 73.1% engaged in heavy episodic drinking or “binge” drinking (defined as 3 or more standard drinks per occasion in this study). Following pregnancy awareness, 4 (19.0% of the 21 women responding to the question) indicated that they continued to drink, with three of the four reporting alcohol consumption at least once per week. About one-quarter of women (24 or 25.8%) reported using marijuana or hashish prior to pregnancy recognition, with roughly half of these women using daily. After pregnancy recognition, 12 (15.0%) reported continued use at any level. Women reporting tobacco use decreased from 29 (30.9%) prior to pregnancy recognition to 11 (13.9%) following pregnancy recognition. Number of cigarettes smoked per day prior to pregnancy recognition averaged 9.3 ± 7.6, and after pregnancy recognition averaged 5.6 ± 4.8. Exposure to secondhand smoke was reported by 42.6% of women prior to and 27.5% after pregnancy recognition.

A total of 20 children were classified with FASD. Prevalence estimates according to the classification of FASD used in the CoFASP study are shown in Table 3. ARND was identified in 14/20 (70.0%) and pFAS in 6/20 (30.0%). No cases of FAS were identified. The overall minimum FASD prevalence in this sample was 41.0 per 1000 or 4.1%. None of the parents or guardians of the 20 children identified were aware prior to the study that their child had an FASD.

## 4. Discussion

The present study used active case ascertainment to establish a minimum FASD prevalence in an AI reservation-based population. The estimated prevalence of 41.0 per 1000 or 4.1% was consistent with the range found in the parent study in four communities in different regions of the United States (11.0–50.0 per 1000 or 1.1–5.0%). In the general population prevalence study, of 222 cases of FASD identified, 27 (12.2%) were classified as FAS, 104 (46.8%) were classified as pFAS and 91 (41.0%) were classified as ARND. However, in the present sample, the highest proportion of FASD cases were classified as ARND (14/20 or 70.0%).

The study was undertaken not only to obtain a prevalence estimate for resource allocation purposes but to identify children affected by FASD and their families for referral for intervention and treatment. Early intervention is crucial to avoid secondary disabilities [25] and to maximize benefit from available treatment and support, yet children with FASD are frequently under-diagnosed or misdiagnosed [26]. At the study venue, expert diagnostic services for FASD had not previously been available. Locally relevant data was needed to harness the power of the community itself to address the issue.

Previous studies have found varying patterns of drinking among AIAN populations, in general higher abstention rates than in the general population, and a higher likelihood of engaging in heavy episodic alcohol consumption when drinking [4,5,6,7,27]. In characterizing our sample population, there appeared to be a strong social norm to engage in heavy episodic alcohol consumption when drinking. Average drinks per drinking day were remarkably similar for men and women (5.87 vs. 5.81). Previous studies within the same community have documented similar levels of binge drinking and that most women do not drink at all [10,11].

No cases of FAS were identified in our study which may reflect resilience within the community that should be further explored or potential differential expression of the manifestations of prenatal alcohol exposure in this population. Conversely, it is also possible that sampling bias led to under-ascertainment of children with full blown FAS. By the same token, nearly twice the proportion of cases of ARND were identified in this AI sample (70.0%) compared to the four regions in the parent study (41.0%). This could have been due in part to the fact that there was no screening tier for growth in the AI sample. As growth deficiency is not a criterion for ARND, it is possible that in the parent study, more children who would have met criteria for ARND were screened out on growth.

### 4.1. Limitations and Strengths

This study had a number of limitations. Less than a quarter of eligible children participated, and participants were self-selected. The small number of child-caregiver dyads recruited and the small number of cases in each category of FASD naturally involve more uncertainty compared to larger studies. Prenatal exposures, including alcohol, were self-reported or, in the case of a collateral interview, reported by the current caregiver. It was not possible to validate maternally reported exposure data by biomarkers or laboratory measures. The study was limited to single assessments at a narrow age-range and it cannot be discounted that deficits may develop at a later time. Some children could not be classified with or without an FASD due to incomplete data collection. It is possible that these data were not missing at random.

The study was conducted in an AI community. There are substantial barriers to conducting research in AIAN communities that may influence findings. Among many AIAN communities there is a lack of trust in research and particularly in **non**-CBPR (community based participatory research) studies where methods are imposed from the outside. This may be due to the legacy of colonialism, the Mission system in our area, historical trauma, and continuing discrimination as well as negative experiences with research [28,29,30,31]. Optimal studies are developed within a community using locally-relevant, culturally congruent motivations and methods. While our study was not “optimal”, it responded to a community prioritized concern, modified the protocol where possible and proceeded with respect. Flexibility was built into the assessments employed, and some were minimally modified.

There are 573 federally recognized AIAN tribes in the United States with remarkably diverse histories, traditions, living conditions, disease burdens, and cultures. These characteristics may influence the reliability of assessment tools [32]. Importantly, not all standardized measures used in this study had been validated among AIAN populations much less this AI population. In particular, the neurobehavioral testing battery had not been specifically validated for AI communities and may be vulnerable to cultural differences. There is an unknown contribution of culture to our findings. Investment is needed in development of valid, culturally congruent measures.

Our study also had strengths. There was community support for the study and recruitment was conducted by respected, trusted community members. The study was broadly publicized using culturally congruent methods. Transportation and refreshments were provided. We sought to minimize reporting bias by assuring participants of confidentiality, explaining Tribal IRB approval, and by carefully training staff. Furthermore, the expertise and experience of the investigators and examiners from the parent study were strengths. Active case-ascertainment was also a strength, and the use of the standardized protocol and measures developed for the parent study allows for comparison with other communities.

### 4.2. Local Relevance

As a result of the present study, we identified a need within the community for FASD-related prevention, services, treatment, and support. We further identified families affected by FASD allowing us to facilitate the goals of an ongoing access to interventions project to link families and children to needed services. The study helped raise awareness of FASD and currently available services. It provided opportunities to decrease stigma and increased the capacity of the local clinic and the community to address FASD.

### 4.3. Implications for Public Health

Our findings contribute to the understanding of FASD among minority populations, specifically AIAN populations. In addition to providing information helpful for clinical resource allocation, this study raises questions about community resilience and the effect of culture. Future research into community-specific risk and protective factors is warranted.

## 5. Conclusions

The estimated minimum prevalence of FASD among a sample of reservation-based American Indians in Southern California was 41.0 per 1000 or 4.1%. This estimate is consistent with the range of estimates from four regional general population samples in the CoFASP study. Of the 20 cases classified in this sample, no cases of FASD had been previously identified. These findings may not be generalizable to all AIAN populations given the heterogeneity of American Indian communities.

## Figures and Tables

**Figure 1 ijerph-16-02179-f001:**
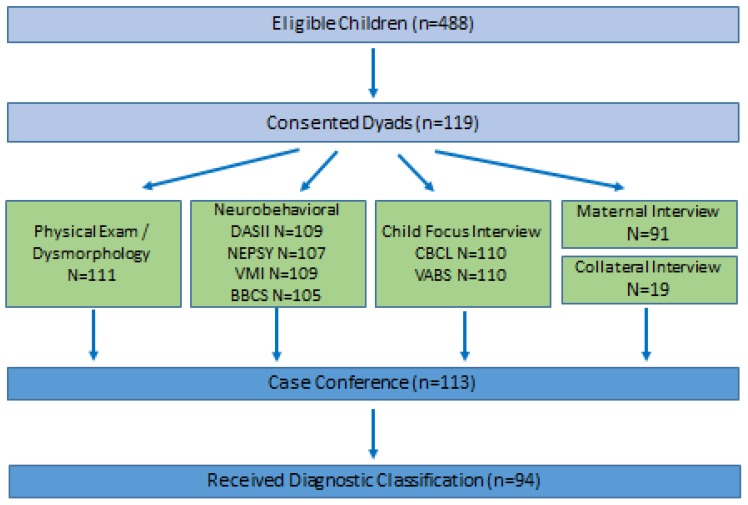
Study Flowchart.

**Table 1 ijerph-16-02179-t001:** Characterization of Sample.

	Mean ± SD or Number (%)
**Children**	
Age, years	6.61 ± 0.91
Sex, female	47 (50.0)
**Mothers**	
Current Age, years	32.6 ± 5.3
Parity	3.04 ± 1.24
Marital Status	
Married	29 (36.3)
Unmarried, living with partner	28 (35)
Widowed	1 (1.3)
Divorced	5 (6.3)
Separated	4 (5.0)
Single	13 (16.3)
Highest Education Level	
Some High School ^1^	6 (7.5)
High School Graduate ^2^	25 (31.3)
Some College or Two-year Degree	41 (51.2)
Bachelor’s Degree	6 (7.5)
Graduate or Professional School	2 (2.5)
Any alcohol during pregnancy	28 (29.8)
Met CoFASP alcohol exposure criteria	18 (19.1)
**Fathers**	
Age (at child’s birth), years	28.0 ± 6.1
Number drinks typically consumed on drinking day	5.87 ± 12.79
Number days per month consumed alcohol	13.5 ± 10.0
**Primary Caregiver**	
Mother	80 (85.1)
Father	1 (1.1)
Grandmother	6 (6.4)
Grandfather	1 (1.1)
Aunt	3 (3.2)
Adoptive, Step, or Foster Parent	3 (3.2)

^1^ High School is 9th through 12th grade ^2^ The majority of students are 17–19 years at graduation.

**Table 2 ijerph-16-02179-t002:** Prenatal Exposures.

	Mean (SD) or Number (%)
**Alcohol during pregnancy, Prior to pregnancy awareness**	28 (29.8)
Women drinking at least once per week	16
Drinks per drinking occasion	5.81 ± 4.71
Women binge drinking when drinking	19
**Alcohol during pregnancy, After pregnancy recognition**	4 (19.0)
Women drinking at least once per week	3
**Marijuana or hashish during pregnancy**	
Any, Prior to pregnancy recognition	24 (25.8)
Any, After pregnancy recognition	12 (15.0)
**Tobacco**	
Any, Prior to pregnancy recognition	29 (30.9)
Amount, Cigarettes per smoking day, Prior to pregnancy recognition	9.25 ± 7.60
Any, After pregnancy recognition	11 (13.9)
Amount, Cigarettes per smoking day, After pregnancy recognition	5.63 ± 4.82
**Second-hand tobacco smoke**	
Any, Prior to pregnancy recognition	40 (42.6)
Any, After pregnancy recognition	22 (27.5)

**Table 3 ijerph-16-02179-t003:** FASD Diagnostic Categories.

Diagnostic Category	*N*	Prevalence Estimate Among 488 Eligible Children
ARND	14	28.7 per 1000 (2.9%)
PFAS	6	12.3 per 1000 (1.2%)
FAS	0	0
Total FASD	20	41.0 per 1000 (4.1%)

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
