# Peer review of "The Prevalence of Fetal Alcohol Spectrum Disorders in An American Indian Community"

_ijerph, 2019, doi:10.3390/ijerph16122179_

Round 1
Reviewer 1 Report
Your paper is very well written and is an important addition to the literature on FASD surveillance, FASD in the AIAN community, and working with tribal communities. I only have a few comments.
I appreciate that you added lines 224-229 discussing the effects of the screening tier for growth. That information is important for FASD surveillance in general.
Lines 243 is the only part that I strongly suggest changing. I don’t understand the use of the word “partly” here. If used, I think the other part needs to be stated. Otherwise, the sentence may be misinterpreted as blaming the victim. There is also a negative historical legacy from past research in the Native American community that you might mention.
I was left wondering if some children with FAS or FASD might have left the community for foster care/adoption. If you have any information on that please considering adding.
Author Response
Our sincere thanks to the reviewers for their helpful comments regarding the manuscript titled “The prevalence of fetal alcohol spectrum disorders in an American Indian community”.
We have attempted to address all suggestions (please see blue font in attachment).
If there is anything else you would like to bring to our attention, please do not hesitate to contact us.
Best regards,
Annika Montag and co-authors

Reviewer 2 Report
please have a look in the attached file

Author Response

(The authors gave the same response as above.)

Round 2
Reviewer 2 Report
please have a look in the attached file

Author Response
The authors thank the reviewer for their helpful suggestions. Please see attached file for response.
